# Task-Oriented Koopman-Based Control with Contrastive Encoder

**Xubo Lyu**
Simon Fraser University
`xlv@sfu.ca`

**Hanyang Hu**
Simon Fraser University
`hha160@sfu.ca`

**Seth Siriya**
University of Melbourne
`ssiriya@student.unimelb.edu.au`

**Ye Pu**
University of Melbourne
`ye.pu@unimelb.edu.au`

**Mo Chen**
Simon Fraser University
`mochen@cs.sfu.ca`

**Abstract:** We present task-oriented Koopman-based control that utilizes end-to-end reinforcement learning and contrastive encoder to simultaneously learn the Koopman latent embedding, operator, and associated linear controller within an iterative loop. By prioritizing the task cost as the main objective for controller learning, we reduce the reliance of controller design on a well-identified model, which, for the first time to the best of our knowledge, extends Koopman control from low to high-dimensional, complex nonlinear systems, including pixel-based tasks and a real robot with lidar observations. Code and videos are available here.

**Keywords:** Learning and control, Koopman-based control, Representation learning

## 1 Introduction

Robot control is crucial and finds applications in various domains. Nonlinear and linear control are two primary approaches for robot control. Nonlinear control [1, 2, 3] is suitable for complex systems when a good nonlinear dynamical model is available. But such a model is not easy to obtain and the nonlinear computation can be sophisticated and time-consuming. Linear control [4, 5, 6] is relatively simple to implement and computationally efficient for systems with linear dynamics, but can exhibit poor performance or instability in realistic systems with highly nonlinear behaviors. Based on Koopman operator theory [7], Koopman-based control [8, 9, 10, 11] offers a data-driven approach that reconciles the advantages of nonlinear and linear control to address complex robot control problems. It transforms the (unknown) nonlinear system dynamics into a latent space in which the dynamics are (globally) linear. This enables efficient control and prediction of nonlinear systems using linear control theory.

Numerous studies have been done on Koopman-based control and they typically follow a two-stage model-oriented process [11, 12, 13]. The first stage is to identify a Koopman model – that is, a globally linear model – from system data, which involves finding a Koopman operator and its associated embedding function to represent linearly evolving system dynamics in the latent space. Classical methods use matrix factorization or solve least-square regression with pre-defined basis functions, while modern methods leverage deep learning techniques [14, 12, 11, 13, 15, 16], such as deep neural networks (DNNs) and autoencoder frameworks, to enhance Koopman model approximation. In the second stage, a linear controller is designed over the latent space based on the Koopman model. Various optimal control methods for linear systems, including Linear Quadratic Regulator (LQR) [17, 13, 12] and Model Predictive Control (MPC) [18, 19, 20, 16], have been employed.

The model-oriented approach in the aforementioned works prioritizes Koopman model accuracy for prediction rather than control performance. While it allows the model to be transferred and reused across different tasks, it has certain limitations. Firstly, it involves a sequential two-stage process, where the performance of the controller is highly dependent on the prediction accuracy of the Koop-

7th Conference on Robot Learning (CoRL 2023), Atlanta, USA.

man model. Thus, slight prediction inaccuracies of the learned model can significantly degrade the subsequent control performance. Secondly, even if the model is perfect, the cost function parameters for the linear controller (e.g. Q and R matrices in LQR controller) need careful manual tuning in both *observed and latent space* in order to have good control performance. These challenges are particularly pronounced in problems with high-dimensional state spaces, thus restricting the applicability of Koopman-based control to low-dimensional scenarios.

**Contributions**. In this paper, we propose a task-oriented approach with a contrastive encoder for Koopman-based control of robotic systems. Unlike existing works that prioritize the Koopman model for prediction, our task-oriented approach emphasizes learning a Koopman model with the intent of yielding superior control performance. To achieve this, we employ an end-to-end reinforcement learning (RL) framework to *simultaneously* learn the Koopman model and its associated linear controller over latent space within a single-stage loop. In this framework, we set the minimization of the RL task cost to be the primary objective, and the minimization of model prediction error as an auxiliary objective. This configuration has the potential to alleviate the aforementioned limitations: (1) RL optimization provides a dominant, task-oriented drive for controller update, reducing its reliance on accurate model identification, (2) manual tuning of cost function parameters is unnecessary as they can be learned implicitly along with the controller in the end-to-end loop.

We adopt a contrastive encoder as the Koopman embedding function to learn the linear latent representation of the original nonlinear system. In contrast to the commonly-used autoencoder, the contrastive encoder is demonstrated to be a preferable alternative, delivering latent embedding that is well-suited for end-to-end learning, especially in high-dimensional tasks such as pixel-based control. To design the Koopman controller, we develop a differentiable LQR control process to be the linear controller for the Koopman latent system. This controller is gradient-optimizable, allowing us to integrate it into the end-to-end RL framework and optimize its parameters through gradient backpropagation. We empirically evaluate our approach across various simulated tasks, demonstrating superior control performance and accurate Koopman model prediction. We compare our approach with two-stage Koopman-based control and pure RL, and deploy it on a real robot.

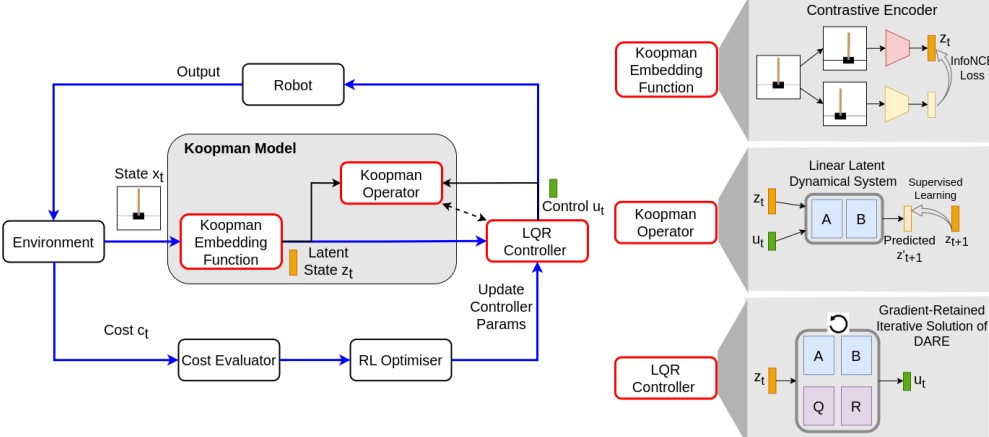

Figure 1: Overview of our method. We adopt an end-to-end RL framework to simultaneously learn a Koopman model and its associated controller. The Koopman model includes a contrastive encoder as the embedding function and a linear matrix as the operator. The Koopman controller is integrated into the loop as a differentiable LQR controller that allows for gradient-based updates. We optimize the entire loop by considering the task cost as the primary objective and incorporating contrastive and model prediction losses as auxiliary objectives.

## 2   Related Work

**Koopman-Based Control.** B.O. Koopman [7] laid the foundation for analyzing nonlinear systems through an infinite-dimensional linear system via the Koopman operator. Subsequent works pro-

posed efficient computation algorithms such as dynamical mode decomposition (DMD) [21, 22] and extended DMD (EDMD) [23, 24] to approximate the Koopman operator from observed time-series data. Recent research has expanded the Koopman operator theory to controlled systems [9, 25], and explored its integration with various control techniques such as LQR [26], MPC [20, 18, 27, 19], pulse control [28]. The emergence of deep learning has further enhanced the learning of Koopman embedding and operator using neural networks and autoencoders [14, 15], enabling their integration with optimal control techniques [11, 12, 16].

**Contrastive Representation Learning.** Contrastive representation learning has emerged as a prominent approach in self-supervised learning in computer vision and natural language processing [29, 30, 31, 32, 33, 34, 35], where it employs an encoder to learn a latent space where the latent representation of similar sample pairs are proximate while dissimilar pairs are distant. Recent works have extended contrastive learning to RL for robot control. Particularly, CURL [36] learns a visual representation for RL tasks by matching embeddings of two data-augmented versions of the raw pixel observation in a temporal sequence. The use of a contrastive encoder on RL enables effective robot control directly from high-dimensional pixel observations.

**Relations to Our Work.** Our work falls into the realm of using deep learning for Koopman-based control. In contrast to existing two-stage approaches [11, 12] involving model identification and controller design, we propose a single-stage, end-to-end RL loop that simultaneously learns the Koopman model and controller in a task-oriented way. We also draw inspiration from the use of contrastive encoder [36], and specifically tailor it as a Koopman embedding function for nonlinear systems with physical states and pixel observations. Our approach enhances the Koopman-based control to be used in high-dimensional control tasks beyond traditional low-dimensional settings.

## 3 Problem Formulation

Consider an optimal control problem over a nonlinear, controlled dynamical systems

$$\min_{\mathbf{u}_{0:T-1}} \sum_{k=0}^{T-1} c\left(\mathbf{x}_k, \mathbf{u}_k\right) \quad \text{subject to } \mathbf{x}_{k+1} = \mathbf{f}\left(\mathbf{x}_k, \mathbf{u}_k\right), \tag{1}$$

where the state $\mathbf{x}$ evolves at each time step $k$ following a dynamical model $\mathbf{f}$. We aim to find a control sequence $\mathbf{u}_{0:T}$ to minimize the cumulative cost $c(\mathbf{x}_k, \mathbf{u}_k)$ over $T$ time steps. Koopman operator theory [7, 9] allows the lifting of original state and input space $\mathbf{x} \in \mathbf{X}$ and $\mathbf{u} \in \mathbf{U}$ to a *infinite-dimensional* latent embedding space $\mathbf{z} \in \mathbf{Z}$ via a set of scalar-valued embedding functions $g : (\mathbf{X}, \mathbf{U}) \to \mathbf{Z}$, where the evolution of latent embedding $\mathbf{z}_k = g(\mathbf{x}_k, \mathbf{u}_k)$ can be globally captured by a linear operator $\mathcal{K}$, as shown in Eq. (2).

$$\mathcal{K}g\left(\mathbf{x}_k, \mathbf{u}_k\right) \triangleq g\left(\mathbf{f}\left(\mathbf{x}_k, \mathbf{u}_k\right), \mathbf{u}_{k+1}\right) \tag{2}$$

Identifying the Koopman operator $\mathcal{K}$ as well as the embedding function $g$ is the key to Koopman-based control. In practice, $\mathcal{K}$ is often approximated using a finite-dimensional matrix $\mathbf{K}$, and the choice of $g$ is typically determined through heuristics or learning from data. Recent research [11, 12] has employed neural networks $\psi(\cdot)$ to encode state $\mathbf{x}$, and define the Koopman embedding function $g(\mathbf{x}, \mathbf{u}) = [\psi(\mathbf{x}) \quad \mathbf{u}]$. Correspondingly, $\mathbf{K}$ is decoupled into state and control components, denoted by matrices $\mathbf{A}$ and $\mathbf{B}$, to account for $\psi(\mathbf{x})$ and $\mathbf{u}$ respectively. This results in a linear time-invariant system with respect to $\psi(\mathbf{x})$ and $\mathbf{u}$ in Eq. (3), facilitating linear system analysis and control synthesis.

$$\mathbf{K}g(\mathbf{x}_k, \mathbf{u}_k) = [\mathbf{A} \quad \mathbf{B}]^\top [\psi(\mathbf{x}_k) \quad \mathbf{u}_k] = \mathbf{A}\psi(\mathbf{x}_k) + \mathbf{B}\mathbf{u}_k = \psi(\mathbf{x}_{k+1}) \tag{3}$$

The goal of Koopman-based control is to identify the Koopman operator $\mathbf{K} = [\mathbf{A} \quad \mathbf{B}]^\top$, the embedding function $\psi(\mathbf{x})$ as well as a linear controller $\mathbf{u} = \pi(\mathbf{x})$ to minimise the total task cost.

## 4 Method: Task-Oriented Koopman Control with Contrastive Encoder

### 4.1 Contrastive Encoder as Koopman Embedding Function

Deep neural networks are extensively employed as flexible and expressive nonlinear approximators for learning Koopman embeddings in a latent space. Inspired by the success of contrastive learning,

we adopt a contrastive encoder to parameterize the embedding function $\psi(\cdot)$. Specifically, for each state $\mathbf{x}_i$ in the state set $\mathcal{X} = \{\mathbf{x}_i \mid i = 0, 1, 2, ...\}$, we create its associated query sample $\mathbf{x}_i^q$ and a set of key samples $\mathbf{x}_i^k$ that include positive and negative samples $\mathbf{x}_i^+$ and $\{\mathbf{x}_j^- \mid j \neq i\}$. $\mathbf{x}_i^+$ is generated by using different versions of augmentations on $\mathbf{x}_i$, while $\{\mathbf{x}_j^- \mid j \neq i\}$ are generated by applying similar augmentations for all the other states: $\mathcal{X} \backslash \{\mathbf{x}_i\} = \{\mathbf{x}_j \mid j \neq i\}$.

Following [32, 36], we use two separate encoders $\psi_{\theta_q}$ and $\psi_{\theta_k}$ to compute the latent embeddings: $\mathbf{z}_i^q = \psi_{\theta_q}(\mathbf{x}_i^q)$, $\mathbf{z}_i^+ = \psi_{\theta_k}(\mathbf{x}_i^+)$ and $\mathbf{z}_j^- = \psi_{\theta_k}(\mathbf{x}_j^-)$. We compute the contrastive loss $\mathcal{L}_{\text{cst}}$ over RL data batch $\mathcal{B}$ based on Eq. (4) to update encoders parameters $\theta_q$, $\theta_k$ and $W$, which is a learnable parameter matrix to measure the similarity between the query and key samples. Two encoders $\psi_{\theta_q}$ and $\psi_{\theta_k}$ are used for contrastive loss computation, but eventually only $\psi_{\theta_q}$ serves as the Koopman embedding function, and we simplify its notation as $\psi_\theta$. We use $\mathbf{t} = (\mathbf{z}, \mathbf{u}, \mathbf{z}', r, d)$ to denote a tuple with current and next latent state $\mathbf{z}, \mathbf{z}'$, action $\mathbf{u}$, reward $r = -c$ and done signal $d$.

$$\mathcal{L}_{\text{cst}} = \mathbb{E}_{\mathbf{t} \sim \mathcal{B}} \log \left( \frac{\exp(\mathbf{z}_i^{q\top} W \mathbf{z}_i^+)}{\exp(\mathbf{z}_i^{q\top} W \mathbf{z}_i^+) + \sum_{j \neq i} \exp(\mathbf{z}_i^{q\top} W \mathbf{z}_j^-)} \right) \tag{4}$$

Different encoder structures and augmentation strategies are required to handle system states depending on how they are represented. For pixel-based states, we adopt convolutional layers as the encoder structure and apply random cropping for augmentation [32, 36]. For physical states, we utilize fully connected layers as the encoder structure and augment the states by adding uniformly distributed, scaled random noise as defined in Eq. (5). $\mathbf{x}^{|\cdot|}$ refers to element-wise absolute of $\mathbf{x}$.

$$\Delta \mathbf{x} \sim \mathrm{U}(-\eta \mathbf{x}^{|\cdot|}, \eta \mathbf{x}^{|\cdot|}); \quad \mathbf{x}^+ = \mathbf{x} + \Delta \mathbf{x} \tag{5}$$

## 4.2 Linear Matrices as Koopman Operator

The Koopman operator describes linear-evolving system dynamics over the latent embeddings and can be represented by a matrix $\mathbf{K}$. Following Eq. (3), we decompose $\mathbf{K}$ into two matrices $\mathbf{A}$ and $\mathbf{B}$, representing the state and control coefficient of a linear latent dynamical system.

$$\mathbf{z}_{k+1} = \mathbf{A}\mathbf{z}_k + \mathbf{B}\mathbf{u}_k \tag{6}$$

To learn $\mathbf{A}$ and $\mathbf{B}$, we optimise a model prediction loss $\mathcal{L}_m$, which is described by Mean-Squared-Error (MSE) as defined in Eq. (7). $\hat{\mathbf{z}}_{k+1}$ is the latent embedding obtained through contrastive encoder at $k + 1$ step. It supervises the predicted latent embedding at the $k + 1$ step from Eq. (6).

$$\mathcal{L}_{\text{m}} = \mathbb{E}_{\mathbf{t} \sim \mathcal{B}} \|\hat{\mathbf{z}}_{k+1} - \mathbf{A}\mathbf{z}_k - \mathbf{B}\mathbf{u}_k\|^2; \quad \hat{\mathbf{z}}_{k+1} = \psi_\theta(\mathbf{x}_{k+1}) \tag{7}$$

## 4.3 LQR-In-The-Loop as Koopman Linear Controller

Given Koopman embeddings $\mathbf{z} = \psi_\theta(\mathbf{x})$ and its associated linear latent system parameterized by $\mathbf{K} = [\mathbf{A} \quad \mathbf{B}]^\top$ shown in Eq. (6), Koopman-based approaches allow for linear control synthesis over latent space $\mathbf{Z}$. Formally, consider the infinite time horizon LQR problem in Koopman latent space that can be formulated as Eq. (8) where $\mathbf{Q}$ and $\mathbf{R}$ are state and control cost matrices. In practice, we choose to represent $\mathbf{Q}$ and $\mathbf{R}$ as diagonal matrices to maintain their symmetry and positive definiteness. The

---

**Algorithm 1:** Iterative solution of DARE

1: Set the total number of iterations $M$
2: Prepare current $\mathbf{A}, \mathbf{B}, \mathbf{Q}, \mathbf{R}$; initialise $\mathbf{P}_M = \mathbf{Q}$.
3: **for** $m = M, M - 1, M - 2, ..., 1$ **do**
4:     $\mathbf{P}_m = \mathbf{A}^\top \mathbf{P}_{m+1} \mathbf{A} - \mathbf{A}^\top \mathbf{P}_{m+1} \mathbf{B} (\mathbf{R} + \mathbf{B}^\top \mathbf{P}_{m+1} \mathbf{B})^{-1} \mathbf{B}^\top \mathbf{P}_{m+1} \mathbf{A} + \mathbf{Q}$
5: **end for**
6: Compute linear gain:
    $\mathbf{G} = (\mathbf{B}^\top \mathbf{P}_1 \mathbf{B} + \mathbf{R})^{-1} \mathbf{B}^\top \mathbf{P}_1 \mathbf{A}$
7: Generate optimal control for latent embedding $\mathbf{z}$:
    $\mathbf{u}^* = -\mathbf{G}\mathbf{z}$

---

LQR latent reference, denoted as $\mathbf{z}_{\text{ref}}$, can be obtained from $\psi(\mathbf{x}_{\text{ref}})$ if $\mathbf{x}_{\text{ref}}$ is provided. $\mathbf{z}_{\text{ref}}$ can also be set to $\mathbf{0}$ if $\mathbf{x}_{\text{ref}}$ is not available. This is particularly useful in cases where the LQR problem does not have an explicit, static goal reference, such as controlling the movement of a cheetah.

$$\min_{\mathbf{u}_{0:\infty}} \sum_{k=0}^{\infty} \left[ (\mathbf{z}_k - \mathbf{z}_{\text{ref}})^\top \mathbf{Q} (\mathbf{z}_k - \mathbf{z}_{\text{ref}}) + \mathbf{u}_k^\top \mathbf{R} \mathbf{u}_k \right] \quad \text{subject to} \quad \mathbf{z}_{k+1} = \mathbf{A}\mathbf{z}_k + \mathbf{B}\mathbf{u}_k, \tag{8}$$

Solving the LQR problem in Eq. (8) involves solving the Discrete-time Algebraic Riccati Equation (DARE). One way this can be done is to take a standard iterative procedure to recursively update the solution of DARE until convergence, as shown in Algo. 1. In practice, we find performing a small number of iterations, typically $M < 10$, is adequate to obtain a satisfactory and efficient approximation for the DARE solution. Thus, we build a LQR control policy $\pi_{\text{LQR}}$ over Koopman latent embedding $\mathbf{z}$ while dependent on a set of parameters $\mathbf{A}, \mathbf{B}, \mathbf{Q}, \mathbf{R}$, as described by Eq. (9).

$$\mathbf{u} \sim \pi_{\text{LQR}}(\mathbf{z}|\mathbf{G}) \triangleq \pi_{\text{LQR}}(\mathbf{z}|\mathbf{P}_1, \mathbf{A}, \mathbf{B}, \mathbf{R}) \triangleq \pi_{\text{LQR}}(\mathbf{z}|\mathbf{A}, \mathbf{B}, \mathbf{Q}, \mathbf{R}) \tag{9}$$

Together with $\mathbf{z} = \psi_\theta(\mathbf{x})$, Eq. (9) implies that the Koopman control policy $\pi_{\text{LQR}}$ is differentiable with respect to the parameter group $\mathbf{\Omega} = \{\mathbf{Q}, \mathbf{R}, \mathbf{A}, \mathbf{B}, \psi_\theta\}$ over the input $\mathbf{x}$. Therefore, this process can be readily used in our gradient-based, end-to-end RL framework. During learning, we follow Algo. 1 to dynamically solve an LQR problem (8) at each step $k$ with current parameters $\mathbf{\Omega}$ to derive a control $\mathbf{u}_k$ for the robot. To optimize the controller $\pi_{\text{LQR}}$ towards lowering the task-oriented cost, we adopt a well-known RL algorithm, soft actor-critic (SAC) [37], to maximize the objective of Eq. (10) via off-policy gradient ascent over data sampled from batch buffer $\mathcal{B}$. $Q_1, Q_2$ are two Q-value approximators used in SAC. In principle, any other RL algorithms can also be utilized.

$$\mathcal{L}_{\text{sac}} = \mathbb{E}_{\mathbf{t} \sim \mathcal{B}} \left[ \min_{i=1,2} Q_i(\mathbf{z}, \mathbf{u}) - \alpha \log \pi_{\text{sac}}(\mathbf{u} \mid \mathbf{z}) \right]; \quad \mathbf{z} = \psi_\theta(\mathbf{x}) \tag{10}$$

## 4.4 End-to-End Learning for Koopman Control

We summarise the previous discussions and present the end-to-end learning process for task-oriented Koopman control with contrastive encoder, as illustrated in Fig. 1 and Algo. 2. We repeatedly collect batches of trajectory data from task environment $\mathcal{E}$ and utilise three objectives to update the parameter group $\mathbf{\Omega} = \{\mathbf{Q}, \mathbf{R}, \mathbf{A}, \mathbf{B}, \psi_\theta\}$ at each iteration. We take the RL task loss $\mathcal{L}_{\text{sac}}$ from Eq. (10) to be the primary objective to optimize all parameters in $\mathbf{\Omega}$ for achieving better control performance on the task. We use contrastive learning $\mathcal{L}_{\text{cst}}$ and model prediction $\mathcal{L}_{\text{m}}$ losses from Eq. (4) and Eq. (7) as two auxiliary objectives to regularise the parameter learning. $\mathcal{L}_{\text{cst}}$ is used to update $\psi_\theta(\cdot)$ to ensure a contrastive Koopman embedding space, while $\mathcal{L}_{\text{m}}$ is used to update $\mathbf{A}, \mathbf{B}$ to ensure an accurate Koopman model in the embedding space.

---

**Algorithm 2:** End-to-End Learning for Koopman Control

1: Initialise Koopman control parameters $\mathbf{Q}, \mathbf{R}, \mathbf{A}, \mathbf{B}, \psi_\theta$
2: Reset task environment $\mathcal{E}$.
3: Initialise a data replay buffer $\mathcal{D}$.
4: **for** iteration $\eta = 0, 1, 2, ...$ **do**
5:     Collect new roll-outs $\tau_\eta$ from $\mathcal{E}$ by running policy $\pi_{\text{LQR}}$ following **Algo. 1**, and save $\tau_\eta$ to $\mathcal{D}$.
6:     Sample a batch of data $\mathcal{B}$ from $\mathcal{D}$.
7:     Compute $\mathcal{L}_{\text{sac}}, \mathcal{L}_{\text{cst}}, \mathcal{L}_{\text{m}}$ based on $\mathcal{B}$.
8:     Update $\mathbf{\Omega} = \{\mathbf{Q}, \mathbf{R}, \mathbf{A}, \mathbf{B}, \psi_\theta\}$ based on $\mathcal{L}_{\text{sac}}$.
9:     Update $\psi_\theta$ and $\mathbf{A}, \mathbf{B}$ based on $\mathcal{L}_{\text{cst}}$ and $\mathcal{L}_{\text{m}}$ respectively.
10: **end for**

---

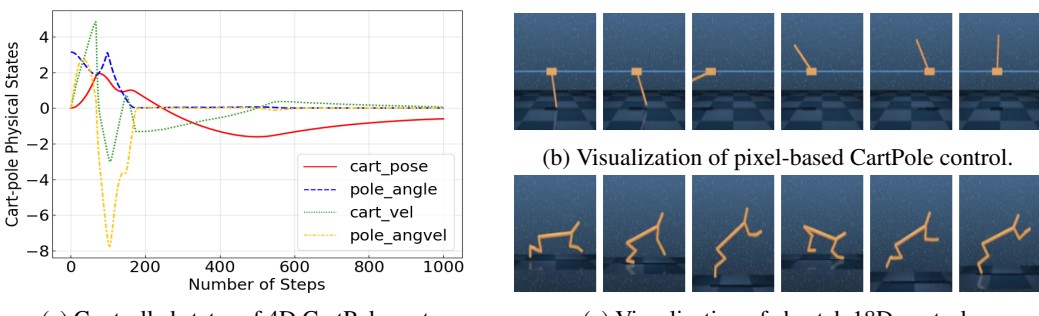

(a) Controlled states of 4D CartPole system

(b) Visualization of pixel-based CartPole control.

(c) Visualization of cheetah 18D control.

Figure 2: Dynamical system behaviors obtained by learned Koopman controller.

# 5 Simulation Results

We present simulated experiments to mainly address the following questions: (1) Can our method achieve desirable Koopman control performance for problems involving different state spaces with different dimensionalities? (2) Are we able to obtain a well-fitted globally linear model in the latent space? For all control tasks, we assume the true system models are unknown.

## 5.1 Task Environments

We include three robotic control tasks with varying dimensions in their state and control spaces from DeepMind Control Suite Simulator [38]: (1) **4D CartPole Swingup.** The objective of this task is to swing up a cart-attached pole that initially points downwards and maintain its balance. To achieve this, we need to apply proper forces to the cart. This task has 4D physical states of cart-pole kinematics as well as 1D control. (2) **18D Cheetah Running.** The goal of this task is to coordinate the movements of a planar cheetah to enable its fast and stable locomotion. It has 18D states describing the kinematics of the cheetah's body, joints, and legs. The 6D torques are used as the control to be applied to the cheetah's joints. (3) **Pixel-Based CartPole Swingup.** The CartPole swingup task with the third-person image as state.

## 5.2 Result Analysis

We report the results in Fig. 2, 3, 4 to demonstrate the effectiveness of our method. Fig. 2 showcases dynamical system behaviors by running a learned Koopman controller. The state evolution and temporal visual snapshots of the three tasks illustrate the successful control achieved by our method. Fig. 3 shows the Koopman controller's performance by comparing its evaluation cost with the reference cost at various learning stages. The reference cost, obtained from [36], is considered the optimal solution to the problem. All experiments are tested over 5 random seeds. Across all three tasks, our method can eventually reach within 10% of the reference cost and continues to make further progress. This indicates our method is generally applicable to both simple, low-dimensional systems and very complex systems involving high-dimensional physical and pixel states.

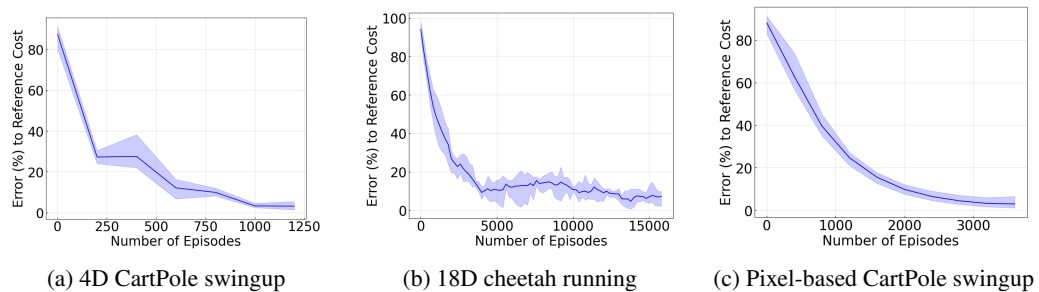

(a) 4D CartPole swingup     (b) 18D cheetah running     (c) Pixel-based CartPole swingup

Figure 3: Mean and standard deviation of error between reference cost and our controller cost during learning.

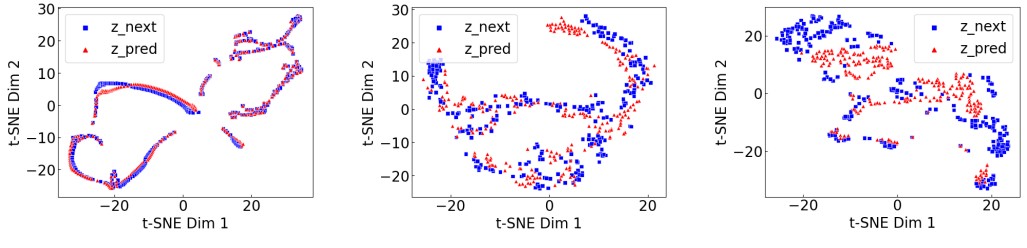

(a) 4D CartPole swingup with mean model error: $2.72 \times 10^{-3}$    (b) 18D cheetah running with mean model error: $8.10 \times 10^{-2}$    (c) Pixel-based CartPole with mean model error: $3.71 \times 10^{-1}$

Figure 4: Distribution maps of 2D data points projected via t-SNE from latent trajectories. **z_next** denotes true trajectories while **z_pred** denotes predicted trajectories using learned Koopman model.

Fig. 4 shows the Koopman model's prediction accuracy in the latent space. We employ t-SNE [39] to project the latent trajectories from a 50D latent space onto a 2D map for improved visualization. The plot in Fig. 4 shows the true and predicted states from trajectories consisting of 1000 steps. Significant overlapping and matching patterns are observed in the distribution of the data points for the 4D CartPole and 18D cheetah systems. This, plus the model prediction error, indicates the potential of utilizing a globally linear latent model to capture the state evolution in both simple and highly complex nonlinear systems. However, for pixel-based CartPole control, the projected states do not perfectly match, suggesting difficulties in accurately modeling the pixel space. Nevertheless, our method still achieves good control performance, even with slight modeling inaccuracies. This highlights the advantage of our approach where the controller is less affected by the model.

## 6 Comparison with Other Methods

### 6.1 Ours vs. Model-Oriented Koopman Control

We compare our method with model-oriented Koopman control (MO-Kpm), which often requires a two-stage process of Koopman model identification and linear controller design. We compare with the most recent work [12] and conduct analysis through the 4D CartPole-swingup task.

**Controller More Robust to Model Inaccuracy.** Table. 1 presents the performance of the Koopman controller under varying levels of Koopman model accuracy. MO-Kpm experiences a rapid increase in total control cost with slightly increasing model error. In contrast, our method demonstrates superior and consistent control performances, indicating its better control quality as well as less

| Model Error | MO-Kpm | | TO-Kpm (Ours) | |
|---|---|---|---|---|
| | Total Cost | Cost Var | Total Cost | Cost Var |
| $\sim 10^{-4}$ | -188.10 | - | **-872.18** | - |
| $\sim 10^{-3}$ | -107.67 | 42.75% | **-846.88** | **2.90%** |
| $\sim 10^{-2}$ | -64.32 | 40.27% | **-784.01** | **7.42%** |

Table 1: Total control cost and its variation under different levels model error using MO-Kpm and our method.

dependency on the model's accuracy. This advantage arises from designing the controller primarily based on task-oriented costs rather than relying heavily on the model. Thus, our method is applicable not only to low-dimensional systems but also to complex and high-dimensional scenarios, such as the cheetah and pixel-based CartPole, where MO-Kpm cannot obtain a reasonable control policy.

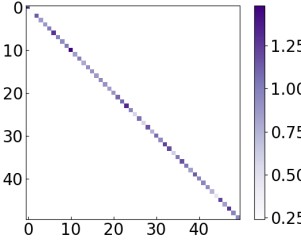

Figure 5: Learned Q matrix

| MO-Kpm | Total Cost |
|---|---|
| $\mathbf{Q}_1 = \mathbf{Diag}(84.12, 62.07, 65.79, 0.04, 0.04, 0, ...)$ | -188.10 |
| $\mathbf{Q}_2 = \mathbf{Diag}(0.01, 10, 10, 0.01, 0.01, 0, 0, ...)$ | -109.23 |
| $\mathbf{Q}_3 = \mathbf{Diag}(10, 60, 60, 0.01, 0.01, 0, 0, ...)$ | -70.65 |
| $\mathbf{Q}_4 = \mathbf{Diag}(10, 60, 60, 10, 10, 0.1, 0.1, ...)$ | -124.80 |
| TO-Kpm (Ours) | Total Cost |
| $\mathbf{Q}$ is shown in Fig. 5 | **-846.88** |

Table 2: Manually tuned and learned Q matrices for latent LQR, and their associated control costs.

**Automatic Learning of Q Matrix in Latent Space.** One major challenge of MO-Kpm is the difficulty in determining the state weight matrix $\mathbf{Q}$ for the latent cost function (Eq. (8)), especially for latent dimensions that may not have direct physical meanings. This challenge can lead to poor control performance, even when the identified model is perfect. Table. 2 compares the control costs obtained from several manually tuned $\mathbf{Q}$ matrices under the best-fitted Koopman model ($10^{-4}$ level) with the learned $\mathbf{Q}$ using our method. Our approach enables automatic learning of $\mathbf{Q}$ over latent space and achieves the best control performance.

### 6.2 Ours vs. CURL

We compare our method with CURL [36], a model-free RL method that uses a contrastive encoder for latent representation learning and a neural network policy for control.

**System Analysis using Control Theory.** Our method differs from CURL in that we learn a linear Koopman model, whereas CURL does not. The presence of a Koopman model (parameterized by $\mathbf{A}, \mathbf{B}$ in Eq. 6) allows us to analyze the system using classical control theory and provides insights for optimizing the controller design. For the CartPole system, we perform stability analysis on both the 50D latent and the 4D true systems, and draw the pole-zero plots in Fig. 6. We find that the learned system demonstrates the same inherent instability as the true system, with the true system's poles accurately reflected in the poles of the latent system (overlapping blue and red dots).

We analyze the controllability of the learned latent system and find its matrix rank to be 6, which indicates that a latent dimension of 50 results in excessive uncontrollable states. Using this information, we apply our method with a lower-dimensional 6D latent space and it is able to maintain the same control and model performance. Further decreasing the latent dimension to 4 leads to degraded control performance, suggesting that the controllability matrix rank is valuable for controller design. This demonstrates the benefit of having an interpretable representation of the state space.

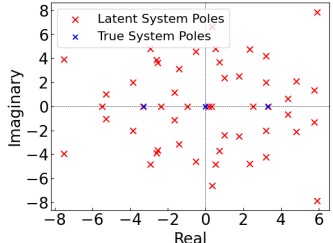

Figure 6: Pole-zero plot of true and learned latent CartPole systems.

| Latent System Dimensions | Total Cost | Model Error |
|---|---|---|
| $\mathbf{Dim}(\mathbf{Z}) = 50$ | -846.88 | $7.76 \times 10^{-3}$ |
| $\mathbf{Dim}(\mathbf{Z}) = \mathrm{rank}(W_{\mathbf{Z}}) = 6$ | **-834.18** | $\mathbf{6.3 \times 10^{-3}}$ |
| $\mathbf{Dim}(\mathbf{Z}) = 4$ | -253.80 | $5.4 \times 10^{-2}$ |
| CURL Control Performance | -841 | - |

Table 3: Our method achieves comparable control cost to CURL while providing more interpretable information about the system.

# 7 Zero-Shot Sim-to-Real Evaluation

We deploy our algorithm trained from the Gazebo simulator to the turtlebot3 burger ground robot. We use 2D Lidar measurements as well as the odometry information as observation, and the linear LQR policy generates the linear and angular velocities as control. We aim to control the robot to navigate through a narrow curved path without any collisions. We directly transfer the trained policy (which is trained with only 40 episodes and each episode contains approximately 700 steps) to the hardware without any fine-tuning, demonstrating the applicability of our approach to a real robot.

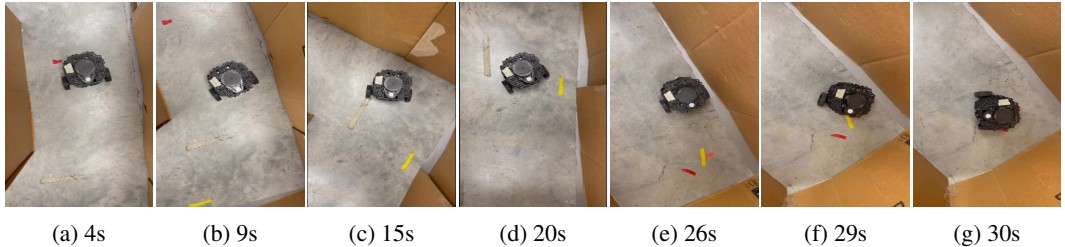

|  |  |  |  |  |  |  |
|---|---|---|---|---|---|---|
| (a) 4s | (b) 9s | (c) 15s | (d) 20s | (e) 26s | (f) 29s | (g) 30s |

Figure 7: Snapshots of real robot curved trajectory at different time stamps (seconds).

# 8 Conclusion and Limitations

In this work, we propose task-oriented Koopman-based control with a contrastive encoder to enable simultaneous learning of the Koopman embedding, model, and controller in an iterative loop which extends the application of Koopman theory to high-dimensional, complex systems.

**Limitation:** End-to-end RL sometimes suffers from poor data efficiency. Therefore, it can be beneficial to leverage an identified model from model-oriented approaches to derive a linear controller to initialise end-to-end RL and improve efficiency. Furthermore, the method has only been validated through simulations and needs hardware deployment for a more practical evaluation.

**Acknowledgments**

We thank reviewers for their invaluable feedback and express a special appreciation to our lab mate Rakesh Shrestha for his help on the hardware setup. This project received support from the NSERC Discovery Grants Program, the Canada CIFAR AI Chairs program, and Huawei Technologies Canada Co., Ltd. Ye Pu's research was supported by the Australian Research Council (DE220101527).

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

# Appendix

## Comparison with Other MBRL Methods

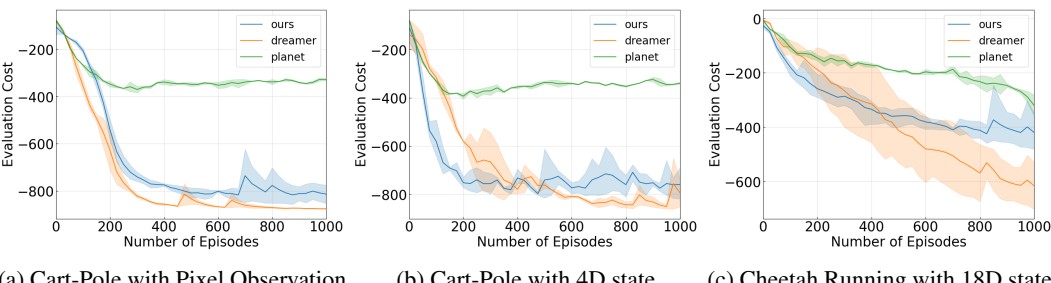

(a) Cart-Pole with Pixel Observation     (b) Cart-Pole with 4D state     (c) Cheetah Running with 18D state

Figure 8: Comparison with well-known model-based RL methods.

To benchmark our approach against established model-based RL methods, we select two widely recognized methods: PlaNet [40] and Dreamer (Version 2) [41]. This comparison spans three tasks in the main paper. Remarkably, across all three tasks, our method demonstrates a clear superiority over PlaNet in terms of both data efficiency and peak performance. Additionally, our method achieves competitive performance with Dreamer-v2 in two Cart-Pole experiments, along with comparable data efficiency in cheetah running tasks. It's important to note that this comparable performance is achieved while our method learns a globally linear model, whereas Dreamer employs a much larger nonlinear network to approximate a world model. Therefore, our approach achieves a substantial reduction in both computational demands and structural complexity while not compromising control performance too much. It is also worth noting that our approach stands out from commonly used Model-Based RL methods as it's rooted in Koopman theory, offering the advantage of enabling control theory analysis for the system.

## Comparison with Other Encoders and Losses

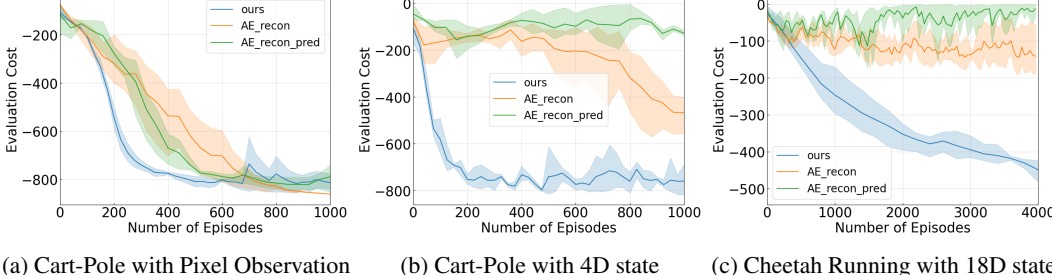

(a) Cart-Pole with Pixel Observation     (b) Cart-Pole with 4D state     (c) Cheetah Running with 18D state

Figure 9: Comparison with autoencoder and varying losses.

We perform experiments to validate our selection of the contrastive embedding function. To achieve this, we replace the contrastive encoder with a canonical autoencoder (AE), a commonly used method for learning condensed representations from high-dimensional observations. Specifically, we utilize the AE along with two types of loss functions: one involving only reconstruction loss, and another involving a combination of reconstruction and one-step prediction loss. Importantly, we keep all other aspects of our method unchanged. Results are based on 3 random seeds.

As shown in Figure. 9, our approach achieves comparable control performance with the use of AE. This aligns with the notion that autoencoder excels in reconstructing and representing pixel-based observations. However, when tasks involve non-pixel observations, such as normal states, our approach still maintains significant control efficiency and performance, while the AE-based structure struggles to learn a useful policy even with ample data. Particularly, we observed that the AE with only reconstruction loss slightly outperforms the one employing the combined loss, but still falls

short of achieving the performance obtained by our method using the contrastive encoder. These results provide validation for our choice of contrastive embedding function within our approach.

**Ablations Study of Hyper-parameters**

We undertook an ablation study involving key parameters of the LQR solving iteration and Koopman embedding dimension. Results are the mean over 3 random seeds and summarized in Table. 4.

We found that our approach remains robust regardless of the specific number of iterations used for the LQR solution, as long as it falls within a reasonable range. This suggests that achieving a certain level of precision in solving the Riccati Equation contributes positively to both policy and linear model learning. We also studied the impact of varying the latent embedding dimensions for the encoder. Our findings indicate that using a smaller dimension that aligns with the encoder's intermediate layers yields consistently good results, while excessively increasing the embedding dimension (d=100) can diminish control performance. One reason could be the reduced approximation capabilities of the encoder due to inappropriate high latent dimension. It might also be because the latent state becoming overly sparse and failing to capture crucial information for effective control.

| | | LQR iteration | | | Latent Dimension | | |
|---|---|---|---|---|---|---|---|
| | | iter=3 | iter=5* | iter=10 | d=30 | d=50* | d=100 |
| cartpole pixel | control cost | -863 | -873 | -835 | -848 | -873 | -813 |
| | model error | 0.075 | 0.058 | 0.269 | 0.165 | 0.058 | 0.038 |
| cartpole state | control cost | -751 | -762 | -769 | -777 | -762 | -667 |
| | model error | 0.00047 | 0.00031 | 0.00042 | 0.00043 | 0.00031 | 0.0002 |
| cheetah run | control cost | -436 | -464 | -448 | -477 | -464 | -235 |
| | model error | 0.732 | 0.862 | 0.842 | 0.653 | 0.862 | 0.046 |

Table 4: Ablation results of final cost and model-fitting error regarding LQR solving iteration and Koopman latent state dimension. The asterisk refers to the parameter used in the main paper.

**Interpretable Q Matrix in Latent Space.**

One key distinction of our method from CURL [36] is the utilization of a structured LQR policy in the latent space. In Fig. 10, we illustrate that the LQR policy parameters, especially the Q matrix, can capture the relative significance weights of latent embedding and their relationship to the original pixel states. We take the pixel-based Cart-Pole task as an example. The larger diagonal elements in the learned Q matrix correspond to visual patches that contain the CartPole object, which provides interpretable information that captures useful latent information related to the CartPole object's area in the image

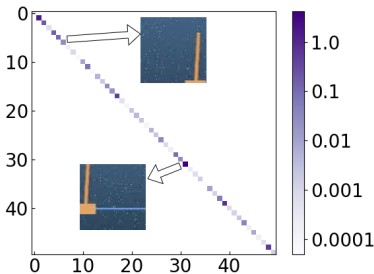

Figure 10: Relations of learned weights in latent **Q** matrix and original pixel states

