# OpenReview forum: "Task-Oriented Koopman-Based Control with Contrastive Encoder"
_robot-learning.org/CoRL/2023/Conference — CoRL 2023 Oral_

### Official Review · Reviewer_kvw1 · 2023-07-18

**Confidence:** 3
**Originality:** Good
**Technical Quality:** Good
**Clarity Of Presentation:** Good
**Impact:** 3

**Recommendation:**

Weak Accept: I recommend accepting the paper, but will not argue for my recommendation if the majority of other reviewers have a different opinion.

**Review:**

the paper is clearly written and easy to follow and understand the algorithmic setup. I also think the approach is interesting and to a certain degree novel in the sense that it stitches together different components coming from optimal control and ML. The end to end approach is interesting especially also the structure around learning the contrastive embedding and linearising the system this way. However the evolution of the method is not mature enough yet for publication. besides of the evaluation being purely in simulation and on fairly easy tasks the paper also lacks comparison to more baselines. Other MBRL baselines, especially ones that combine learning approaches with optimal control should be compared against, it would be interesting to compare in terms of actual task performance to see whether there is improved performance and also whether there is a benefit in sample efficiency.  The paper only compares to total control cost which is a bit harder to interpret in terms of how useful the approach is.

**Quality Of The Limitations Section:**

Limitations are addressed clearly

**Questions For Rebuttal:**

see above

**Robotics Focus:**

Relevant but unlikely to deploy to hardware in near future

**Summary Of Paper:**

The paper proposes a Model Based RL approach based on Koopman control.The paper proposes a contrastive learning approach to learn the koopman embedding as well as the linear dynamical system within the latent space. this setup can then allow to use classical control techniques to solve the control problem. The paper is clearly written and easy to understand the different parts of the paper and the algorithms. The experimental evaluation is fully in simulation on fairly simple control suite tasks. The results show that the proposed approach reaches near optimum performance on the tasks as well as decent predicting of the state space.

**Summary Of Recommendation:**

given the lack of more realistic experimental evaluation I believe this paper is not ready for publication at CoRL yet.

---

> ### Author Response · Authors · 2023-08-09
> **Response to Reviewer kvw1**
>
> Thank you for your valuable comments. We answer the questions as follows.
> > Q1: Lack of other MBRL baselines especially those combined with optimal control
>
> A1: We recognize the need for comparisons with other model-based RL approaches. Currently, we're conducting experiments to contrast the Koopman-based method with two established techniques: PlaNet[1] and Dreamer[2]. Preliminary results highlight clear data efficiency advantages over these methods. We'll compile a comprehensive report in an upcoming appendix and promptly share it.
>
> [1] Hafner, Danijar, et al. "Learning latent dynamics for planning from pixels." International conference on machine learning. PMLR, 2019.
>
> [2] Hafner, Danijar, et al. "Mastering atari with discrete world models." arXiv preprint arXiv:2010.02193 (2020).
> > Q2: Lack of more realistic experimental
>
> A2: Thanks for pointing this out. We are currently in the process of conducting realistic experiments, and once they are complete, we will make sure to upload the updated information. Your patience is greatly appreciated.

---

### Official Review · Reviewer_XqmK · 2023-07-18

**Confidence:** 3
**Originality:** Very Good
**Technical Quality:** Good
**Clarity Of Presentation:** Very Good
**Impact:** 4

**Recommendation:**

Strong Accept: I recommend accepting the paper and will argue for my recommendation even if other reviewers hold a different opinion.

**Review:**

## Strengths and Weaknesses
### Strengths
- well written and evaluated paper
- an interesting idea
- good results including on pixel based tasks

### Weaknesses
- no real world evaluation
- no clear indication for such attempts
- potentially very difficult to scale to complex physical tasks, e.g. like soft object manipulation

**Quality Of The Limitations Section:**

Limitations are addressed clearly

**Questions For Rebuttal:**

- This method seems scalable to real world scenarios, why are there no physical world evaluations. With those, I will be happy to increase my score.

**Robotics Focus:**

Highly relevant to robotics but no hardware experiments

**Summary Of Paper:**

This work proposes a novel method for task-oriented koopman based control. The proposed method utilises end-to-end reinforcement learning and contrastive encoder to simultaneously learn the Koopman latent embedding operator and associated linear controller. Overall, the paper is written very well but it is evaluated purely in simulation. Having said that, the provided evaluation and analysis is very comprehensive and overall pleasant to read. The proposed method has the potential to be scaled to physical systems and therefore I recommend it for acceptance. A disclaimer here is that I am not an expert on Koopman based control.

**Summary Of Recommendation:**

Overall I enjoyed reading this paper. I did not like that it does not have any physical world experiments but regardless I see the proposed approach to have potential for being impactful.

---

> ### Author Response · Authors · 2023-08-09
> **Response to Reviewer XqmK**
>
> Thank you a lot for your helpful comments, and we answer the questions as follows:
> > Q1: no real-world evaluation
>
> A1: We acknowledge the significance of a real-world assessment and value your openness to boosting our score based on the physical test. We're actively preparing results from a Turtlebot hardware experiment to validate our approach and will share them with you shortly.
> > Q2: no clear indication for such attempt
>
> A2: The indication for our approach is two-fold: (1) Traditional Koopman-based methods struggle with high-dimensional control because of inaccurate model-fitting and tricky tuning of latent control. However, an end-to-end task-oriented optimization such as RL is able to address these problems. Therefore, we structure Koopman-based control into an RL process to expand its capabilities. (2) Koopman approach requires an effective linear latent embedding for a high-dimensional nonlinear system. Given the success of contrastive unsupervised learning in handling high-dimensional visual data, we use a contrastive encoder and associated InfoNCE loss to develop a Koopman embedding.

---

### Official Review · Reviewer_n3f2 · 2023-07-19

**Confidence:** 5
**Originality:** Very Good
**Technical Quality:** Good
**Clarity Of Presentation:** Very Good
**Impact:** 4

**Recommendation:**

Weak Accept: I recommend accepting the paper, but will not argue for my recommendation if the majority of other reviewers have a different opinion.

**Review:**

The paper presents a novel approach to the problem of model-based control using Koopman operators. The concept of the contrastive encoder to model the Koopman embedding function is quite interesting and the interpretability of Koopman operator oriented around tasks rather than prediction error is an approach that has valuable merit in this area.

A major strength of this work is enabling control-based interpretability of learned RL systems. This is often not a property of existing methods and should be highlighted. The ability to further adapt the LQR control problem in the loop is quite novel as a big issue with Koopman control is the ability to appropriately tune controllers based in the embedded space.

However, one key weakness of the work is motivating analysis and experimental validation as to why the contrastive embedding is preferred over other canonical choices (e.g., normal MLP prediction loss). In fact one can perhaps claim that the true benefit is the task-oriented learning achieved with SAC that is key to ensuring the algorithm works well. As an improvement it is key that these kinds of choices have obvious empirical and theoretical backing, otherwise the choice of this approach seems arbitrary. It would benefit the paper to have such evidence that this is the right approach and what the expected improvements are so that one can validate the hypothesis experimentally.

Another minor weakness that can be readily addressed is the lack of an ablation study on the parameters e.g., prediction time T, Koopman embedding dimension, etc. These are valuable as it strengthens the reproducibility of the work and highlights limitations of the method and future directions.

**Quality Of The Limitations Section:**

Additional details required

**Questions For Rebuttal:**

What happens if you let the LQR controller iterate longer?

Does the time horizon of the problem matter?

How stable is the back prop through the LQR controller? Do we get divergence?

What are key limitations of the work regarding implementation and prior knowledge?

Do the specific nonlinearities of the embeddings matter in reconstructing the latent Koopman space or is all the improvement on the contrastive loss?

**Robotics Focus:**

Highly relevant to robotics but no hardware experiments

**Summary Of Paper:**

The paper presents a method for learning control policies based on task-oriented Koopman operators. The main idea being that prioritizing Koopman operator model fidelity around the task has higher performance than simply emphasizing prediction loss of the Koopman operator. The main result would be the ability to construct linear control based on the Koopman operator around the task rather than over the whole state-space. The approach uses a contrastive encoder to construct the Koopman function embedding and integrates an LQR controller learning into the learning loop.

**Summary Of Recommendation:**

The work demonstrates a promising direction within robot learning that should be highlighted. However, the current work lacks sufficient robotics implementation. As such a weak accept is recommended as the work has potential, but requires more thorough robotics results and analysis to make a proper impact.

---

> ### Author Response · Authors · 2023-08-08
> **Response to Reviewer n3f2**
>
> Thank you for the constructive and detailed comments. We provide the responses as follows for the rebuttal questions and paper weaknesses.
> > Q1: What happens if you let the LQR controller iterate longer?
>
> A1: We are uncertain about the intended meaning behind ``let LQR controller iterate longer'', and would greatly appreciate some clarification if possible. However, based on our understanding, there appear to be two possible interpretations for your question and corresponding answers.
>
> (1) If the question pertains to increasing the number of iterations to solve the Riccati equation and generate the LQR control input, this would likely lead to improved convergence and higher accuracy in the LQR solution. However, it's important to consider the trade-off: this approach would extend the time required to solve the LQR equation for each training epoch, resulting in a less efficient overall reinforcement learning process in terms of wall time. Right now we use the number of iterations $M=5$, and we are also experimenting with smaller ($M=3$) and larger numbers ($M=10$) and will upload the results in an appendix file soon.
>
> (2) If the question pertains to whether running a solved LQR controller more times to gather additional roll-out data would significantly impact the Koopman control performance, the influence might be relatively minor.  The current setup involves task episodes with a fixed length of 1000 steps, and our approach incorporates the SAC RL method for off-policy optimization. During each iteration, a consistent batch size ($bz = 128$) of roll-out data is sampled from a replay buffer and subsequently employed for gradient-based policy optimization. Hence, gathering more data using the LQR controller in the replay buffer does not have a direct effect on the learning performance.
> > Q2: Does the time horizon of the problem matter?
>
> A2: We assume the ``time horizon'' here refers to the horizon of formulated LQR problem. In this case, it is less relevant since we are addressing the infinite time horizon LQR problem. We accomplish this by solving the discrete-time algebraic Riccati equation, leading to a unique Riccati solution $P$ and associated control $u(\cdot)$ that span all time steps. We also adjust Eq.8 to appropriately accommodate the notion of an infinite time horizon $\sum_{k=0}^{\infty}$ as follows:
> \begin{equation}
>     \min\_{\mathbf{u}\_{0:\infty}} \sum\_{k=0}^{\infty}\left[\left(\mathbf{z}\_k-\mathbf{z}\_{\mathrm{ref}}\right)^{\top} \mathbf{Q}\left(\mathbf{z}\_k-\mathbf{z}\_{\mathrm{ref}}\right)+\mathbf{u}\_k^{\top} \mathbf{R} \mathbf{u}\_k\right] \text{ subject to }  \mathbf{z}\_{k+1}=\mathbf{A} \mathbf{z}\_k+\mathbf{B} \mathbf{u}\_k.
> \end{equation}
> > Q3: How stable is the back prop through the LQR controller? Do we get divergence?
>
> A: Our current implementation based on  Algo.1 acquires a stable back-propagation process without encountering divergence, and we can proceed with the RL training until its convergence of average return. We believe the proper mathematical calculation of the inverse of a matrix in Algo.1 is of vital importance in obtaining a stable, well-converged LQR solution. The code snippet of Algo.1 is attached here. In line 3, the use of torch.linalg.solve(B, A) provides a much more stable computation of the inverse matrix.
> ```
> for i in reversed(range(T)):
>         # using torch.linalg.solve(B, A) to obtain the solution  of AX = B to avoid direct inverse
>         V_uu_inv_B_trans = torch.linalg.solve(torch.matmul(torch.matmul(B_trans, V[i+1]), B) + R, B_trans)
>         K[i] = torch.matmul(torch.matmul(V_uu_inv_B_trans, V[i+1]), A)
>         k[i] = self._batch_mv(V_uu_inv_B_trans, v[i+1])
>
>         # riccati difference equation, A-BK
>         A_BK = A - torch.matmul(B, K[i])
>         V[i] = torch.matmul(torch.matmul(A_trans, V[i+1]), A_BK) + Q
>         v[i] = self._batch_mv(A_BK.transpose(-2, -1), v[i+1]) + self._batch_mv(Q, goals[:, i, :])
> ```
> > Q4: What are key limitations of the work regarding implementation and prior knowledge?
>
> A4: In practice, one of the key limitations lies in the lack of prior knowledge of an appropriate latent dimensionality for various robotic systems and control tasks, since Koopman theory does not provide insight on the exact dimensionality needed to guarantee linear latent dynamics. However, we have found that selecting a higher latent dimension and using multi-layer convolutional and fully-connected neural network structures as an encoder is effective for learning latent embedding. By maintaining an identical dimensionality ($\text{Dim}(\mathbb{Z})=50$) across all tasks, our learned Koopman model demonstrates the expected linearity (Sec. 5 of the paper).
>
> We continue the remaining response in a new comment.

---

> ### Author Response · Authors · 2023-08-08
> **Response to Reviewer n3f2 (2)**
>
> > Q5: Do the specific nonlinearities of the embeddings matter in reconstructing the latent Koopman space or is all the improvement on the contrastive loss?
>
> A5: If the term "specific nonlinearities of the embeddings" refers to the structure of the encoder, then its impact is less important than the use of contrastive loss. Because different encoder structures, with varying numbers of layers and nodes, can achieve similar approximation abilities to learn latent embedding. We conducted additional experiments using the same encoder but replacing the contrastive loss with a reconstruction loss. We observed poorer control performance, indicating that contrastive loss plays a pivotal role in our setup. We will provide more detailed results in the appendix.
> > Regarding main weakness: one key weakness of the work is motivating analysis and experimental validation as to why contrastive embedding is preferred over other canonical choices (e.g., normal MLP prediction loss). In fact, one can perhaps claim that the true benefit is the task-oriented learning achieved with SAC which is key to ensuring the algorithm works well. As an improvement it is key that these kinds of choices have obvious empirical and theoretical backing, otherwise the choice of this approach seems arbitrary.
>
> Answer: Thank you for your insightful observation. We agree that conducting a comparison between the contrastive encoder and other encoder options (along with their loss functions) holds significant importance, and can demonstrate the necessity of contrastive encoder. We are currently running experiments on comparing contrastive encoder with the standard autoencoder (using reconstruction and prediction loss). Our initial results show that the use of a contrastive encoder and its loss does significantly improve data efficiency and performance of the trained policy. We will upload the results soon and incorporate them into the next version of the paper.
> > Regarding minor weakness: Another minor weakness that can be readily addressed is the lack of an ablation study on the parameters e.g., prediction time T, Koopman embedding dimension, etc. These are valuable as it strengthens the reproducibility of the work and highlights limitations of the method and future directions.
>
> Answer: We agree that having an ablation study of a set of parameters can provide more valuable information. We are running experiments and will report the results on the parameters of the Koopman embedding dimension and the number of iterations on solving the Riccati Equation.
>
> Thank you again for your careful reading and helpful comments. I'm looking forward to your valuable opinion on the replies and further results.

---

### Official Review · Reviewer_rfrp · 2023-07-21

**Confidence:** 3
**Originality:** Good
**Technical Quality:** Good
**Clarity Of Presentation:** Fair
**Impact:** 4

**Recommendation:**

Weak Accept: I recommend accepting the paper, but will not argue for my recommendation if the majority of other reviewers have a different opinion.

**Review:**

The idea of applying koopman control for RL is not original, but the author propose to use RL simultaneously learn the Koopman model and its associated linear controller  instead  of only focusing on the accuracy of the dynamic model. The idea which is simple and effective that makes the model-based controller robust to the dynamic model's ion.

advantages:
- Improves the model-oriented koopman control method's performance. The controller is robust to the learned dynamic model's prediction error.

weakeness:
- The experiment are evaluated on a limited selections of tasks, only 2 simple deepmind control suite tasks and one imaged-based cartpole control task. The dynamics and action dimensions are relatively simple in these tasks.

- Lacks detailed analysis and comparison experiments. For example, this work does not compare the method with general model-based RLs (e.g. dreamver, etc). It neither presents the sample efficiency as it is important for model-based RLs.


**Quality Of The Limitations Section:**

Additional details required

**Questions For Rebuttal:**

Q1 - The RL optimization part is unclear. The primary optimization objective Eq.10 is a typical max-entropy RL objective, while the paper describes the RL optimization as using SAC and it brings confusion. SAC is a typical off-policy actor-critic RL, but here it seems it is optimized in an on-policy manner by using the latent linear dynamics? If so, I think it is just optimized using the basic max-entropy RL instead of SAC. It would be nice to illustrate the RL optimization part more clearly in the appendix.

Q2 - Is it possible to keep matrices A and B fixed during the RL optimization, such that A and B are only optimized for latent dynamics and Q, R is optimized for the task? If jointly optimizing \OMEGA together, dynamic matrices A and B might also get affected.

Q3 - As Koopman control essentially is a model-based solution, some comparison with other model-based RL solutions might be needed to see how beneficial if we take a data-driven Koopman control-based solution than simple model-based RLs.


**Robotics Focus:**

Highly relevant to robotics but no hardware experiments

**Summary Of Paper:**

This paper proposes a novel task-oriented approach using a contrastive encoder for Koopman-based control of robotic systems.

They employ an end-to-end reinforcement learning (RL) framework, simultaneously learning the Koopman model and its associated linear controller over latent space within a single-stage loop. The primary objective is to minimize the RL task cost, with model prediction error minimization as an auxiliary objective.

It use a contrastive encoder to learn the linear latent representation of the nonlinear system. They design the Koopman controller with a differentiable LQR solution process, allowing gradient optimization within the end-to-end RL framework for controller parameter optimization.

The paper presents empirical evaluations through simulations on various tasks and is compared with two-stage Koopman-based control and pure RL methods, offering a comprehensive assessment of its effectiveness.

**Summary Of Recommendation:**

The idea of implementing task performance oriented koopman control is intersting and has certain novelty. However, the evaluation tasks are relatively simple and lacks comparison experiments and analysis. I would recommend a weak accept.

---

> ### Author Response · Authors · 2023-08-07
> **Response to Reviewer rfrp**
>
> Thank you for the careful review and insightful feedback on our manuscript. We truly appreciate your time and effort in evaluating it. We provide the following answers to the questions:
> > Q1: The RL optimization part is unclear. The primary optimization objective Eq.10 is a typical max-entropy RL objective, while the paper describes the RL optimization as using SAC and it brings confusion. SAC is a typical off-policy actor-critic RL, but here it seems it is optimized in an on-policy manner by using the latent linear dynamics? If so, I think it is just optimized using the basic max-entropy RL instead of SAC. It would be nice to illustrate the RL optimization part more clearly in the appendix.
>
> A1: Thanks for the keen observations. Yes, the RL optimization should be a off-policy process. We use SAC in a off-policy way which reuses data from a replay buffer rather than using on-policy data for optimization. The Eq.10 has been updated to be a more clear SAC policy objective: $\mathcal{L}\_{sac}=\mathbb{E}\_{z \sim \mathcal{D}}\left[\min\_{i=1,2} Q\_{i}(z, u)-\alpha \log \pi\_{sac}(u \mid z)\right]$
> , where it computes expected return over data sampled from buffer $\mathcal{D}$. We will also update the method figure to show the process of sampling data from buffer instead of purely from current policy. We will include these updates in later version.
>
> > Q2: Is it possible to keep matrices A and B fixed during the RL optimization, such that A and B are only optimized for latent dynamics and Q, R is optimized for the task? If jointly optimizing $\Omega$ together, dynamic matrices A and B might also get affected.
>
> A2: We did try the option of fixing A and B while only optimizing Q and R, and it leads to much worse results. This is because if A and B are fixed, which means fixing the latent dynamics, then the encoder parameters has to be also fixed to maintain the coherence of the latent embedding space aligned with the latent dynamics. As a result, only very few parameters (including Q and R) can be updated to minimize the task cost. As Q and R are both linear matrices, their capacity for effective function approximation is very constrained. Jointly optimizing $\Omega$ is feasible because A and B are updated based on both Koopman prediction loss $\mathcal{L}\_{m}$ and RL task loss $\mathcal{L}\_{sac}$. Therefore, the joint optimization ensures that A and B consistently maintain latent model prediction ability while also yielding strong control performance.
>
> > Q3: As Koopman control essentially is a model-based solution, some comparison with other model-based RL solutions might be needed to see how beneficial if we take a data-driven Koopman control-based solution than simple model-based RLs.
>
> A3: We agree that some comparison with other model-based RL is necessary. And now we are running experiments to compare the Koopman-based approach with two established model-based methods: PlaNet [1] and Dreamer [2]. The initial observations show evident data efficiency advantages over these alternatives, and we will continue the experiments and summarize the complete results in an appendix file and upload it soon.
>
> Apart from this, it is important to emphasize that, unlike standard MBRL, our key advantage lies in employing a contrastive embedding along with a linear model and classical controller for complex, high-dimensional robot control. This design not only achieves effective control and model learning but also enables interpretability and control theoretic analysis for the latent system (e.g. stability, controllability) and control policy, which is an aspect often under-explored in standard MBRL.
>
> [1] Hafner, Danijar, et al. "Learning latent dynamics for planning from pixels." International conference on machine learning. PMLR, 2019.
>
> [2] Hafner, Danijar, et al. "Mastering atari with discrete world models." arXiv preprint arXiv:2010.02193 (2020).
>
> > Regarding the weakness: The experiment is evaluated on a limited selection of tasks, only 2 simple deepmind control suite tasks and one imaged-based cartpole control task. The dynamics and action dimensions are relatively simple in these tasks.
>
> Answer: We understand this concern. Actually, this approach is primarily designed for addressing the challenges of traditional Koopman-based control using contrastive RL techniques, rather than enhancing RL using Koopman theory. The three examples in this work, although considered as solved by standard RL, (prior to our work, to the best of our konwledge) remain unsolved by Koopman-based approaches due to their strong non-linearity and high dimensionality (especially the 18D cheetah and pixel cartpole examples). Hence, our results offer notable contributions to Koopman control, robustly attaining a globally-linear Koopman model and an associated LQR controller without manual tuning, which opens the door to the interpretability of the system via a vast set of control theoretic analysis tools (Sec. 6).

---

> > ### Comment · Reviewer_rfrp · 2023-08-14
> > **Response to answers**
> >
> > Thanks for the clarification and additional experiments. The additional contents/modifications do answered my questions. Despite the proposed method has relatively worse performance comparing with Dreamer, but relatively seems have better sample efficiency. It would be nice to investigate if the worse performance is caused by the linear latent dynamics or other problems.

---

> > > ### Author Response · Authors · 2023-08-16
> > > **2nd round response to reviewer rfrp**
> > >
> > > We greatly appreciate your response and we are willing to continue investigating more about the root cause of the worse performance and demonstrate it in the paper.  Besides, we are pleased to present the most recent real-world experimental results for your consideration, and include the results in the updated zip file of rebuttal response.
> > >
> > > In this real robot task, we deploy our algorithm trained from Gazebo simulator to the turtlebot3 burger ground robot. We use 2D Lidar measurements as well as the odometry information as observation, and the linear LQR policy generates the linear and angular velocities as control. We aim to control the robot to navigate through a narrow curve path without any collisions. We directly transfer the trained policy (which is trained with only 40 episodes and each episode contains around 700 steps) to the hardware without any fine-tuning, indicating the applicability of our approach on real robot. We will continue train and deploy our approach on more complex tasks.

---

### Decision · Program_Chairs · 2023-08-30

**Decision:**

Accept (Oral)

**Comment:**

This paper presents a task-oriented koopman based control that features end-to-end RL and conrastive encoder to learn the Koopman latent embedding oeperator.

The reviewer found that that the paper presents an interesting and novel ideas in a well-presented manner. The evaluation part is pointed out relatively weak, but the rebuttal better clarifies on this aspect.